# Short-Term Mortality Fluctuations and Longevity Risk-Adjusted Age: Learning the Resilience of a Country to a Health Shock

**Gloria Polinesi** [1,*,†], **Maria Cristina Recchioni** [1,†], **Andrea Rimondi** [2,†] **and Anton Sysoev** [3,†]

1   Department of Economics and Social Sciences, Universitá Politecnica delle Marche, 60121 Ancona, Italy; m.c.recchioni@staff.univpm.it

2   Department of Economics, Mathematics and Statistics, Birkbeck University of London, London WC1E 7HX, UK; andrea.rimondi@outlook.it

3   Department of Applied Mathematics, Lipetsk State Technical University, 398055 Lipetsk, Russia; sysoev_as@stu.lipetsk.ru

*   Correspondence: g.polinesi@staff.univpm.it; Tel.: +39-071-220-7066

†   These authors contributed equally to this work.

**Abstract:** Recent studies have attempted to measure differences in lifestyle quality across the world. This paper contributes to this strand of literature by extending the indicator introduced in Milevsky (2020), i.e., "longevity-risk-adjusted global age" (LRaG age), to deal with the new short-term mortality fluctuation data series freely available from the Human Mortality Database. The new weekly data on mortality allow measuring weekly biological age. The weekly differences between biological and chronological ages across countries were used to assess country resilience to the COVID-19 pandemic in terms of excess mortality and health expenditure. Countries with a biological age lower than the chronological age had a lower excess mortality in 2020–2021 and a lower health expenditure, thus indicating some resilience to the shock of COVID-19.

**Keywords:** longevity risk; Gompertz–Makeham mortality; biological age; short-term mortality

## 1. Introduction

Inspired by new waves of the COVID-19 pandemic, many studies have concentrated on investigating its dramatic consequences on populations around the world and local regional situations. As an example, the study [1] showed that during this period, the highest mortality rate within the last five years among old people was observed. These findings led us to contribute to this area by applying longevity indicators across the Russian and Italian populations for the years 2019 and 2020 [2]. It was shown that the age gap median values decreased in 2020, meaning that the COVID-19 outbreak caused a flattening across all regions between chronological and biological ages.

The current study expanded the method based on the so-called survival analysis of different world regions. The technique uses a set of statistical methods aimed at estimating the probability of a given event's occurrence as a function depending on time [3]. Using a probability density function $f(t)$, it is possible to estimate the cumulative distribution function, $F(t)$, of a certain death at a random time $t$ as:

$$F(t) = P(T < t) = \int_0^t f(u)du,$$

where $T$ represents the random life span and $t$ is the time of death.

Many models that use this method offer estimates in terms of age-dependent mortality rates. Existing studies use parametric and non-parametric techniques in this regard. Mortality rates (also known as hazard rates) are described in a mathematical form that is a

function of age in the former technique. The mortality rate is calculated using a weighted average of the "crude" death rates in the latter technique.

The first category is represented by Gompertz's study [4]. The overall mortality rate in his work was calculated using an age-dependent factor that grows exponentially with chronological age. A further contribution came from William Makeham [5]. He inserted an age-independent term in the preceding framework, giving rise to the Gompertz–Makeham (GM) model. This is one of the foundational concepts in the creation of human mortality theory. Several authors then extended their model to simulate specific forms of mortality rates [6–8]. The paper [9] was a study of parametric functions for mortality modeling that implemented different parametric models.

Milewsky [10] more recently proposed a new concept of age estimation. The author devised a mechanism for inverting GM mortality (hazard) rates, resulting in a new age definition, i.e., the longevity-risk-adjusted global age (L-RaG).

Non-parametric models can be used instead of parametric models to estimate the survival function. They do not make any assumptions about the theoretical distribution of $F(t)$, $t > 0$. Instead, they employ a number of estimators (several of which are often used in clinical trials), the most notable of which are the Kaplan–Meier estimator [11], the Nelson–Aalen estimator [12], and the long-rank test [13].

This study proposes an empirical contribution to existing studies by extending the indicator introduced by Milevsky to obtain L-RaG age estimates across countries between 2017 and 2020 from the new short-term mortality fluctuation data series freely available on the Human Mortality Database website. The new weekly data on mortality allow measuring weekly biological age. The weekly differences between biological and chronological ages across countries were used to assess country resilience to the COVID-19 pandemic in terms of excess mortality and health expenditure.

The paper is organized as follows. In Section 2, the method presented in this work is introduced; in particular, we describe the GM mortality law and the L-RaG age indicator. In Section 3, the main results are compared, and we comment on the differences pre and post the COVID-19 pandemic. Section 4 contains the conclusion.

## 2. The Gompertz–Makeham Mortality Estimation from Weekly Data

The mortality law introduced by Gompertz–Makeham (GM) is defined as a linear model connecting the natural mortality rate and the chronological age $x$. This indicator gives us the number of years a person has been alive. It means that every adult who lives in the country (or region) $i$, for $i = 1, \ldots, N$ can be estimated as:

$$1 - q_x = e^{-\int_x^{x+1} \mu_y dy},\qquad(1)$$

where $\mu_x$ is the total mortality rate and $q_x[i]$, specifically estimated based on country–mortality rate tables, is the effective one-year decrement rate.

To deal with weekly mortality data, we reinterpreted Equation (1) in terms of weekly fractions of a year, $j/n_w$, $j = 1, 2, \ldots, n_w$, where $n_w$ is the number of weeks in the year in question; on average, there are 52 weeks in a year (i.e., $n_w = 52$). To this end, Equation (1) is read as the survival probability at the $j$-th week of the year:

$$1 - q_{x,j} = e^{-\int_x^{x+(j/n_w)} \mu_y dy},\qquad(2)$$

where $q_{x,j}$ denotes the death rate in the interval $(x, x + j/n_w)$. The dynamics of the weekly total hazard rate (wTHR) are analogous to the total hazard rate (THR) as in [10]:

$$\mu_x[i] - \lambda[i] = \begin{cases} h[i]e^{g[i]x} & x < x^* \\ \lambda^* & x \geq x^* \end{cases},\qquad(3)$$

where $\mu_x$ is the weekly total hazard rate and the independent variable $x$ is the age expressed in years plus weeks.

Parameters $\lambda[i]$, $h[i]$, and $g[i]$ are the accidental mortality rate (i.e., Makeham constant), the initial natural mortality rate (INMR), and the mortality growth rate (MGR), respectively. These parameters are specific for each country. The factor $x^*$ is responsible for the limit of the Gompertzian regime, and the condition $\lambda^* > \lambda[i]$ is used as the constant value when chronological ages are equal to $x^*$. These parameters are also known as the species-specific lifespan and the species-specific accidental mortality rate [14]. Model (3) can be linearized as:

$$\ln[\mu_x[i] - \lambda[i]] = \ln(h[i]) + g[i]x, \tag{4}$$

which provides the logarithm of the difference between the weekly total hazard and the accidental death rate as a linear function of age in the GM regime $[x, x^*]$. Indeed, the full Gompertz–Makeham model prescribes a new link among the parameters, i.e.,

$$1 - q_{x,j} = e^{-\int_x^{x+(j/n_w)} (\lambda[i] + h[i]e^{g[i]y}) dy}, \tag{5}$$

so integrating and taking the logarithm of both sides of (5) yields:

$$-\ln(1 - q_{x,j}) = \frac{j}{n_w}\lambda[i] + \frac{h[i]}{g[i]}e^{g[i]x}\left(e^{g[i](j/n_w)} - 1\right), \tag{6}$$

which also reads:

$$\frac{n_w}{j}\ln\left(\frac{1}{1 - q_{x,j}}\right) - \lambda[i] = \frac{h[i]}{g[i]}e^{g[i]x}\frac{n_w}{j}\left(e^{g[i](j/n_w)} - 1\right), \tag{7}$$

that is,

$$\ln\left(\frac{n_w}{j}\ln\left(\frac{1}{1 - q_x[i]}\right) - \lambda[i]\right) = \ln(h[i]) + \ln\left[\left(e^{g[i](j/n_w)} - 1\right)/((j/n_w)g[i])\right] + g[i]x. \tag{8}$$

The model in Equation (8) differs from Milevsky (see [10]) because of the time dependence in the term $\left(e^{g[i](j/n_w)} - 1\right)/((j/n_w)g[i])$. We therefore rewrite Equation (8) to estimate the country-specific GM parameters $\lambda[i]$, $h[i]$, $g[i]$. Specifically, we add and subtract the term $\ln((e^{g[i]} - 1)/g[i])$ on the right-hand side of Equation (8) and use the expansion:

$$\ln\left(\frac{1 + ay}{1 + a}\right) = \ln\left(\frac{1}{1 + a}\right) + ay + o(y), \quad y \to 0, \quad a > 0, \tag{9}$$

where $a = g[i]/2$, $y = j/n_w$. Equation (9) allows us to rewrite (8) as follows:

$$\overbrace{\ln\left(\frac{n_w}{j}\ln\left(\frac{1}{1 - q_x[i]}\right) - \lambda[i]\right)}^{z} \\ = \overbrace{\ln(h[i]) - \ln\left(1 + \frac{1}{2}g[i]\right) + \ln\left[\left(e^{g[i]} - 1\right)/g[i]\right]}^{K_0} + \overbrace{g[i]}^{K_1}(x + j/(2n_w)). \tag{10}$$

It is easy to see that when $j = n_w$, Formula (10) reduces to the one proposed in [10] for small values of $g$, such that $\ln(1 + g[i]/2) \approx g[i]/2$.

We use Equation (10) to estimate the region-specific GM parameters $\lambda[i]$, $h[i]$, $g[i]$, while the global parameters $x^*$ and $\lambda^*$ are estimated using the following linear regression:

$$\ln(h[i]) = L + (-x^*)g[i] + \varepsilon, \tag{11}$$

where, by continuity, $L = \ln \lambda^*$.

The GM mortality law means that this indicator increases linearly the log-mortality rate, which becomes constant when the critical age, $x^*$, is reached. Such a tendency is also known as the compensation law. A linear, negative relationship is therefore assumed to exist between the initial natural mortality (intercept term) $\ln(h[i])$ and the MGR (slope) $g[i]$, as shown in Equation (11). We can derive the relationship between $h[i]$ and $g[i]$, $x^*$ and $\lambda^*$ in (11) by taking the limit of the total hazard rate in Equation (3), which can be considered a continuous function of age, $x$, for $x \to x^*$. Imposing this limit for $x$, which tends to $x^*$ from below and above, we have:

$$h[i]e^{g[i]x^*} = \lambda^* \;\longrightarrow\; h[i] = \lambda^* e^{-g[i]x^*}. \tag{12}$$

We can therefore rewrite $\mu_x[i]$ as follows:

$$\mu_x[i] = \begin{cases} \lambda[i] + \lambda^* e^{g[i](x-x^*)} & x < x^* \\ \lambda[i] + \lambda^* & x \geq x^* \end{cases}. \tag{13}$$

*L-RaG Age Indicator*

In this section, we introduce the L-RaG age Equation (13) with country-specific parameters and a new equation involving global GM parameters.

In line with [10], we assumed that all the differences among countries in Model (13) are included in the longevity-risk-adjusted global age, $\xi(x,i)$. The latter is the age "physically perceived" by an individual with age $x$ in country $i$. To capture this perception, we rephrase the model for the total hazard rate as follows:

$$M_{\xi(x,i)} = \begin{cases} \Lambda + \lambda^* e^{G(\xi(x,i)-x^*)} & x < x^* \\ \Lambda + \lambda^* & x \geq x^* \end{cases}, \tag{14}$$

where $M_{\xi}(x,i)$ is the longevity-risk-adjusted total hazard rate and $\Lambda \geq 0, G \geq 0$ represent the country-level mean of $\lambda[i]$, $g[i]$, respectively (Milevsky (2020) [10]):

$$\Lambda = \frac{1}{N}\sum_{i=1}^{N}\lambda[i], \quad G = \frac{1}{N}\sum_{i=1}^{N}g[i]. \tag{15}$$

The L-RaG age, $\xi(x,i)$, represents the biological or perceived age, i.e., the age estimated by including lifestyle factors such as diet, exercise, and sleeping habits, whose computation involves a mapping from mortality rates to a specific age by inverting the GM mortality law.

For this reason, the L-RaG age must satisfy the compensation law described in (3), accounting for the mean of GM parameters across the countries, i.e., Equations (14) and (15). Setting the country-level longevity-risk-adjusted global hazard rate equal to the total hazard rate, we have:

$$M_{\xi}(x,i) = \mu_x[i]. \tag{16}$$

Solving Equation (14) for the L-RaG age, $\xi(x,i)$, we obtain:

$$\xi(x,i) = x^* + \frac{\ln\left[\lambda[i] - \Lambda + \lambda^* e^{g[i](x-x^*)}\right] - \ln[\lambda^*]}{G}. \tag{17}$$

The validity of $\xi(x,i)$ is related to $\lambda[i] - \Lambda + \lambda^* e^{g[i](x-x^*)} > 0$, and weekly data guarantee that this constraint is satisfied.

Finally, we define the difference, the longevity-risk-adjusted global (dLRaG) indicator, as:

$$dLRaG(x,i) = \xi(x,i) - x. \tag{18}$$

Positive/negative values of this indicator reveal people of age $x$ and living in country $i$ with a perceived age greater/smaller than the chronological age. The temporal dynamics

of this indicator with a negative trend may reveal an improved lifestyle quality, while a positive trend may suggest worse lifestyle conditions.

## 3. Results

We used the weekly LRaG indicator to analyze the evolution of perceived age across countries in the Northern Hemisphere while trying to establish whether this indicator and/or the dLRaG indicator can be applied to measure the well-being of a country along with its resilience to the pandemic shock.

This was performed by showing that the LRaG gap, i.e., the dLRaG indicator, is associated with health expenditure and excess mortality. Specifically, the LRaG gap in 2019 is predictive of excess mortality in 2020–2021, while the LRaG in 2017 and 2020 is associated with health expenditure in 2019. Interestingly, countries with a biological age lower than the chronological age were those with a higher health expenditure and lower excess mortality. These findings suggest that the LRaG gap is a potential tool for measuring a country's resilience to health shocks combined with the size of the elderly population.

### 3.1. Description of the Data

We used the Short-Term Mortality Fluctuation Data series (we thank Dmitri Jdanov, Vladimir M. Shkolnikov, and Ainhoa Alustiza Galarza with the assistance of Carl Boe and Magali Barbieri for providing the dataset freely at https://mortality.org/ (accessed on 21 February 2022)) to estimate LRaG age and, consequently, the dLRaG indicator. The new series was added to the Human Mortality Database triggered by the COVID-19 pandemic of 2019–2020 and the importance of short-term or seasonal mortality fluctuations that are driven by temporary hazards such as epidemics, temperature extremes, natural disasters, and so on. These particular problems tend to affect vulnerable population groups such as elderly people.

We chose these new series motivated by the arguments mentioned in the STMF methodological note (https://www.mortality.org/Public/STMF$\_$-$DOC/STMFNote.pdf (accessed on 21 February 2022)): "Objective and internationally comparable data are crucial to evaluate the political strategies used to address epidemics and other public health crises. Indicators based on disease incidence and fatality as well as on cause-specific mortality are valuable but have important shortcomings that make comparisons across countries and time problematic. In contrast, being able to look at short term fluctuations in all-cause mortality (such as captured by weekly or monthly excess deaths) comprise an important complement to other types of data. Weekly death counts constitute a solid data basis for the most objective and comparable way of assessing the scale of short-term excess mortality across countries and over time.".

The weekly age-specific death rate over the interval $(x, x + a)$ is calculated as:

$$m_y^w(x, x + a) = D_y^w(x, x + a) / (E_y(x, x + a)/52), \tag{19}$$

as a proxy of the death rate $q_{x,a}$. Here, $D_y^w(x, x + a)$ is the number of deaths in a (broader) age interval for week $w$ in year $y$, while $E_y(x, x + a)$ indicates the annual population exposure in age interval $(x, x + a]$ in year $y$. We considered four years, 2017, 2018, 2019, and 2020, and 42 weeks in 2021.

We note that the weekly age-specific death rate is given for age classes: 0–14, 15–64, 65–74, 75–84, and 85-plus. In the following, we identify the class with its middle points; $x$ and weekly age are given by $x + j/n_w$, $j = 1, 2, \ldots, n_w$. We also used the health expenditure available on the OECD.Sta website (https://data.oecd.org/healthres/health-spending.htm (accessed on 21 February 2022)). Specifically, we used the health spending indicator (total/government/compulsory/voluntary, USD/capita, 2020 or the latest available), expressed in USD per capita.

Finally, we used data on excess mortality per 100k people along with excess mortality as a percentage of the annual baseline. The dataset is available thanks to the work of Karlinsky and Kobak (2021) [15], who reported the results of an excess mortality study

that extended to the summer of 2021 and also included middle-income countries. Excess mortality, computed as the mortality that exceeds the baseline level (for details, see https://elifesciences.org/articles/69336 (accessed on 21 February 2022)), depends on infection rates, population demographics, COVID-19 interventions, stressed healthcare systems, and vaccine coverage. We used this cross-sectional data to analyze the relationship between excess mortality due to the shock of the COVID-19 pandemic and pre-existing perceived age over the countries.

### 3.2. Estimation of LRaG Age and dLRaG Indicator

As mentioned in the previous subsection, we used the weekly age-specific death rate, $m_y^w(x, x + j/n_w)$ as the death rate $q_{x,j}$, that is the decremental rate of an individual aged $x$ at the $j$-th week of the year in question. We estimated the GM model using Equations (10) and (11), where $x$, as mentioned above, represents the class considered (i.e., the midpoint of the age class).

The compensation law—the negative relationship between the mortality growth rate and the (log) initial mortality rate—was validated for all age classes and years via the univariate linear regression (11), which allowed estimating the Gompertzian regime limit age, $x^*$ (i.e., the slope), and the non-country-specific part, $\lambda^*$, of the hazard rate (i.e., the intercept). Table 1 shows the estimates of the parameters $x^*$ and $\lambda^*$; it also shows the $p$-value of the slope of the linear regression (11).

**Table 1.** Gompertzian regime limit age, $x^*$, non-country-specific part, $\lambda^*$, of the total hazard rate and $p$-value of the slope in the linear regression (11) for different years.

| Year | $x^*$ | $\lambda^*$ | $p_{value}$ |
|---|---|---|---|
| 2016 | 91 | 0.32 | $4.01 \times 10^{-13}$ |
| 2017 | 88 | 0.27 | $7.42 \times 10^{-11}$ |
| 2018 | 84 | 0.20 | $8.93 \times 10^{-11}$ |
| 2019 | 87 | 0.25 | $1.95 \times 10^{-11}$ |
| 2020 | 81 | 0.16 | $7.86 \times 10^{-9}$ |
| 2021 (42 weeks) | 72 | 0.08 | $5.05 \times 10^{-8}$ |

The temporal changes of the parameters $x^*$ and $\lambda^*$ in Table 1 revealed that the COVID-19 pandemic already affected longevity by reducing the Gompertzian regime limit age $x^*$, while halving a plateau, $\lambda^*$, i.e., reducing the non-country-specific part of the hazard rate. Bearing in mind that individuals who reach advanced age $x^*$ experience an exponentially distributed remaining lifetime with a constant country-dependent hazard rate, our findings suggest that individuals in 2020 will experience the plateau six years earlier than in the pre-COVID period, but with half the risk.

### 3.3. Does the Dynamic Evolution of the dLRaG Indicator Capture Quality of Life?

The temporal changes of the dRaG indicator revealed a transformation in the quality of life. Each panel in Figure 1 shows the map of the dRaG indicator across the world for a given age class and year. Specifically, the panels in the top row relate to the age class 15–64, those in the middle regard age class 65–74, while the panels at the bottom regards the age class 75–84. The panels in a given column relate to the same year; three years are shown: 2017, 2019, and 2020.

The color gray indicates that STMF series are not available. Moving from 2017 to 2020, we observed an increase in the dRaG indicator in northwestern countries and a decrease in northeastern countries. This suggests that the quality of life, as measured by the dLRaG indicator, has been deteriorating in northwestern countries, while improving—despite the COVID-19 pandemic—in northwestern countries.

Such dynamics appeared strongly in age class 15–64 (see also Figure 2 for the years 2017 and 2020), while the evidence weakened for age classes 65–74 and 75–84.

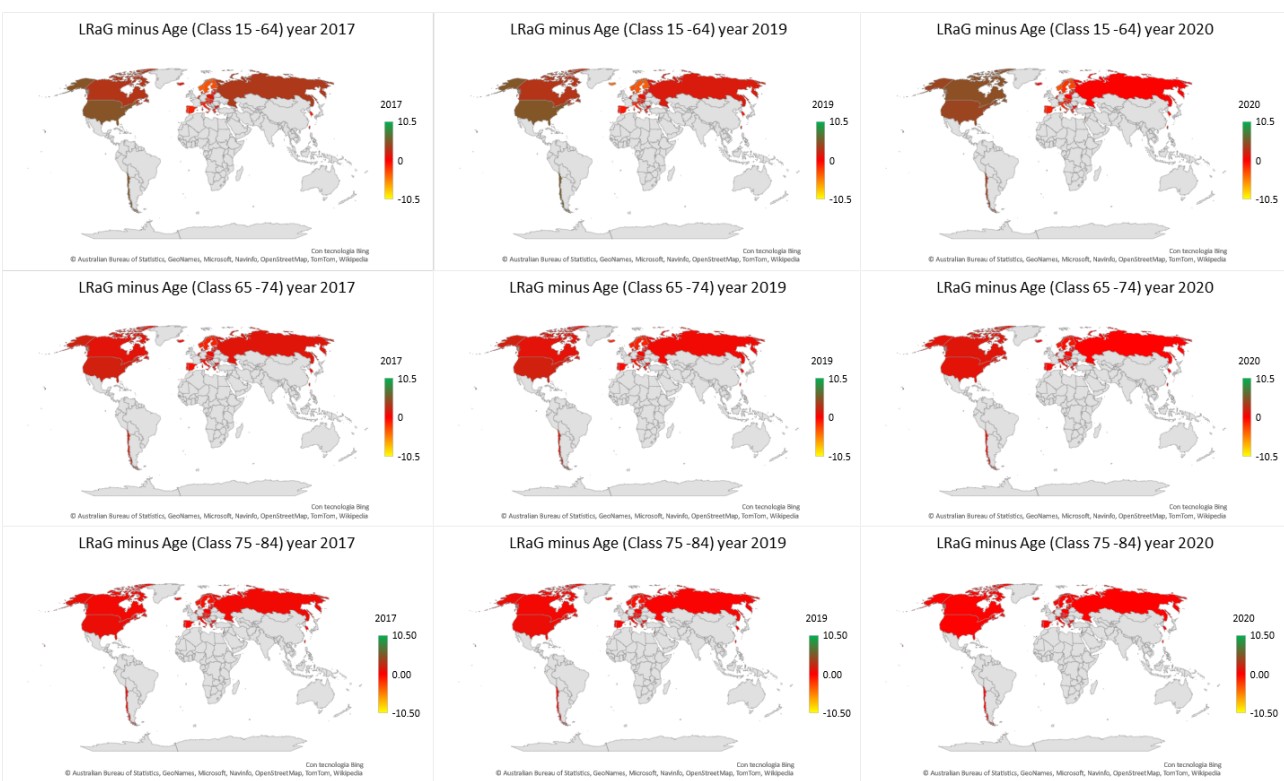

**Figure 1.** dLRaG for three age classes by country. Each panel shows the dLRaG across the countries for a given year and age class.

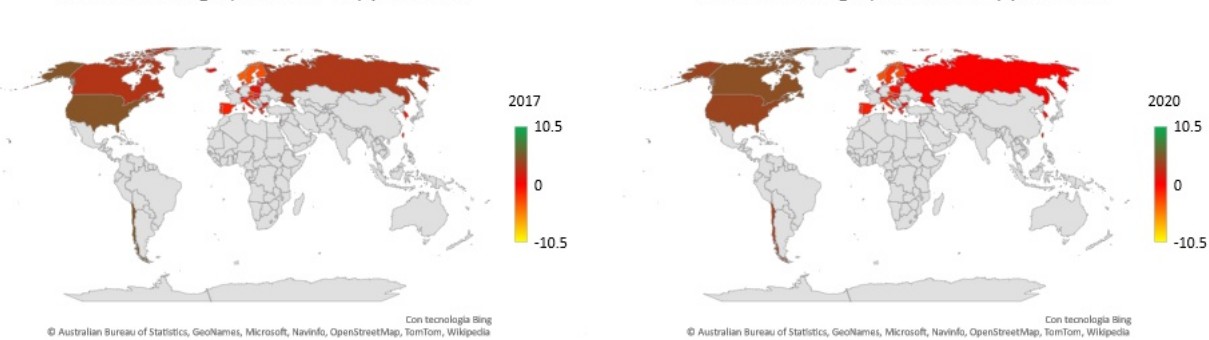

**Figure 2.** dLRaG for age class 15–64 per country and years 2017 and 2020. Each panel shows the dLRaG across the countries.

The difference in the dRaG dynamics of Eastern and western countries in the Northern Hemisphere may rely on differences in the country-specific health expenditure of each country.

We provide empirical evidence of this relationship using the annual health expenditure in 2019 across the countries (data are available at the OECD website https://data.oecd.org/healthres/health-spending.htm (accessed on 21 February 2022)). Specifically, we investigated a multivariate linear regression where the response function is the LRaG gap in 2020 (i.e., the dLRaG indicator in 2020) and the explicative variables are health expenditure in 2019 and the dLRaG indicator in 2017, 2018, and 2019:

$$dLRaG_{i,2020} = \beta_0 + \beta_1 dLRaG_{i,2017} + \beta_2 dLRaG_{i,2018} + \beta_3 dLRaG_{i,2019} + \beta_4 HE_{i,2019} + \epsilon_i, \quad (20)$$

where $i$ denotes the $i$-th country and $\epsilon_i$ is normally distributed noise. The data on health expenditure and gaps in LRaG (i.e., dLRaG) from 2017 to 2020 used to estimate the pa-

rameters of the linear model (20) are shown in Table A1 in Appendix A. We did not consider data relative to the USA since its LRaG dynamics contrasts with the behavior of the remaining countries.

The results of the linear regression were very satisfactory since the multiple R-squared was 0.8437, while the adjusted R-squared was 0.8206 and the *p*-value of the F-statistic was $1.6 \times 10^{-10}$, thus implying empirical evidence to reject the null hypothesis. Furthermore, we cannot reject the hypothesis that residuals are normally distributed, as shown in Table 2, and that the coefficient of the LRaG gap in 2017 and health expenditure in 2019 are significant (see Table 3).

**Table 2.** Normality test for residuals of the multivariate linear regressions (20).

| Test | Statistic | *p*-Value |
|---|---|---|
| Shapiro–Wilk | 0.9835 | 0.8921 |
| Kolmogorov–Smirnov | 0.0822 | 0.9696 |
| Anderson–Darling | 0.2509 | 0.7205 |

**Table 3.** Multivariate linear regression (20). Signif. codes: 0 "***" 0.001 "**", 0.01 "*", 0.05 ".", 0.1 " ", 1.

| Parameter | Estimate | Std. Error | *t* Value | Pr(>\|*t*\|) |
|---|---|---|---|---|
| (intercept) | −0.2061 | 0.1095 | −1.88 | 0.0706 |
| $dLRaG_{2017}$ | 0.3638 | 0.1860 | 1.96 | 0.0609 (·) |
| $dLRaG_{2018}$ | 0.1232 | 0.2534 | 0.49 | 0.6307 |
| $dLRaG_{2019}$ | 0.1426 | 0.1186 | 1.20 | 0.2394 |
| $HE_{2019}$ | 0.0001 | 0.0000 | 2.03 | 0.0526 (·) |

We looked for the best reduced linear model that achieved the best adjusted $R^2$ along with the significance of the coefficients. This resulted in the linear model:

$$dLRaG_{i,2020} = \beta_0 + \beta_1^* dLRaG_{i,2017} + \beta_2^* HE_{i,2019} + \epsilon_i^*, \tag{21}$$

with a multiple R-squared equal to 0.8253 and adjusted R-squared to 0.8133. The *p*-value of the F-statistic was equal to $1.03 \times 10^{-11}$, indicating empirical evidence to reject the null hypothesis of the null coefficients $\beta_1^*$, $\beta_2^*$. Table 4 provides the details of the linear model (21).

**Table 4.** Multivariate linear regression (21). Signif. codes: 0 "***", 0.001 "**", 0.01 "*", 0.05 ".", 0.1 " ", 1.

| Parameter | Estimate | Std. Error | *t* Value | Pr(>\|*t*\|) |
|---|---|---|---|---|
| (intercept) | −0.2113 | 0.1116 | −1.89 | 0.0682 |
| $dLRaG_{2017}$ | 0.6002 | 0.0516 | 11.64 | 0.0000 (***) |
| $HE_{2019}$ | 0.0001 | 0.0000 | 2.00 | 0.0544 (·) |

The residuals of the regression (21) passed the test to be normally distributed, as shown in Table 5.

**Table 5.** Normality test for residuals of the multivariate linear regression (20).

| Test | Statistic | *p*-Value |
|---|---|---|
| Shapiro–Wilk | 0.9649 | 0.3711 |
| Kolmogorov–Smirnov | 0.1046 | 0.8393 |
| Anderson–Darling | 0.4189 | 0.3086 |

Linear Models (20) and (21) showed that the difference between the biological and chronological ages in 2020 across the countries can be predicted using the health expenditure in 2019 and the LRaG gap in 2017–2019 across the countries.

More specifically, bearing in mind that the response function is the LRaG gap in 2020, we observed that an increase of USD 1000 per capita in the health expenditure implied an increase of the LRaG gap of 0.1 year, while an increase of one year in the LRaG gap 2017 implied an increase in the LRaG-2020 gap of 0.6 years.

Interestingly, this was confirmed by the fact that the temporal changes of the LRaG gap (i.e., the dLRaG indicator) were related to the excess mortality recently analyzed, for example, in Islam et al. (2021) [16] and Karlinsky and Kobak (2021) [15]. Specifically, Islam et al. (2021) [16] reported substantial excess mortality in some Eastern European countries and no excess mortality in New Zealand, Norway, or Denmark based on data from 29 high-income countries in 2020. Later, Karlinsky and Kobak (2021) [15] reported the results of an excess mortality study extended to the summer of 2021 and also included middle-income countries.

Our hypothesis is that countries with a favorable LRaG gap (i.e., negative dLRaG indicator) should be resilient to the shock of the pandemic. We addressed this point using two linear models. In the first, Equation (22), the response function is the excess mortality per 100k people up to summer 2021, which we denote with $ME100k_{i,2021}$, where the subscript $i$ refers to the $i$-th country. In the second, Equation (23), the response function is the excess mortality as a percentage of the annual baseline up to summer 2021, which we denote with $MEAB_{i,2021}$, where, once again, the subscript $i$ refers to the $i$-th country. The following linear models were considered:

$$ME100k_{i,2021} = \beta_0 + \beta_1 dLRaG_{i,2017} + \beta_2 dLRaG_{i,2018} + \beta_3 dLRaG_{i,2019} + \beta_4 dLRaG_{i,2020} + \epsilon_i, \quad (22)$$

$$MEAB_{i,2021} = \beta_0^* + \beta_1^* dLRaG_{i,2017} + \beta_2^* dLRaG_{i,2018} + \beta_3^* dLRaG_{i,2019} + \beta_4^* dLRaG_{i,2020} + \epsilon_i^*, \quad (23)$$

where $\epsilon_i$ and $\epsilon_i^*$ are normally distributed noise. The data used to estimate the model parameters are shown in Table A1 in Appendix A. The estimation of Model (22) yielded a multiple R-squared of 0.4012, adjusted R-squared of 0.3125, and p-value of the F-statistic 0.006317, while Model (23) outperformed the previous one with a multiple R-squared of 0.4977, adjusted R-squared of 0.4233, and p-value of the F-statistic 0.0007091. The coefficients are displayed in Table 6. Due the higher values of the variance inflation ratio (VIF) associated with variables $dLRaG_{2017}$ and $dLRaG_{2018}$, 17.14 and 22.27, according to both Models (22) and (23), we eliminated $dLRaG_{2018}$ from the models. Moreover, Figure 3 shows the Akaike information criterion (AIC) values for the stepwise linear regression based on all possible combinations of predictors. Models (22) and (23) reached the minimum value of the AIC only if the three predictors were included in the analysis (AIC = 393 vs. AIC = 230).

**Table 6.** Multivariate linear regressions (22) and (23). Signif. codes: 0 "***", 0.001 "**", 0.01 "*", 0.05 ".", 0.1 " ", 1.

| **Linear Regression (22)** | | | | |
|---|---|---|---|---|
| **Parameter** | **Estimate** | **Std. Error** | ***t* Value** | **Pr(>\|*t*\|)** |
| (intercept) | 138.8954 | 18.7357 | 7.41 | 0.0000 |
| $dLRaG_{2017}$ | 107.5452 | 91.3934 | 1.18 | 0.2496 |
| $dLRaG_{2018}$ | 45.9393 | 122.1673 | 0.38 | 0.7098 |
| $dLRaG_{2019}$ | 77.6427 | 55.0293 | 1.41 | 0.1697 |
| $dLRaG_{2020}$ | −309.4199 | 84.0705 | −3.68 | 0.0010 (**) |

**Table 6.** *Cont.*

| | Linear Regression (23) | | | |
|---|---|---|---|---|
| Parameter | Estimate | Std. Error | *t* Value | Pr(>\|*t*\|) |
| (intercept) | 12.8720 | 1.4451 | 8.91 | 0.0000 |
| $dLRaG_{2017}$ | 8.1244 | 7.0490 | 1.15 | 0.2592 |
| $dLRaG_{2018}$ | 6.0949 | 9.4225 | 0.65 | 0.5232 |
| $dLRaG_{2019}$ | 7.4316 | 4.2443 | 1.75 | 0.0913 ($\cdot$) |
| $dLRaG_{2020}$ | $-27.4200$ | 6.4842 | $-4.23$ | 0.0002 (\*\*\*) |

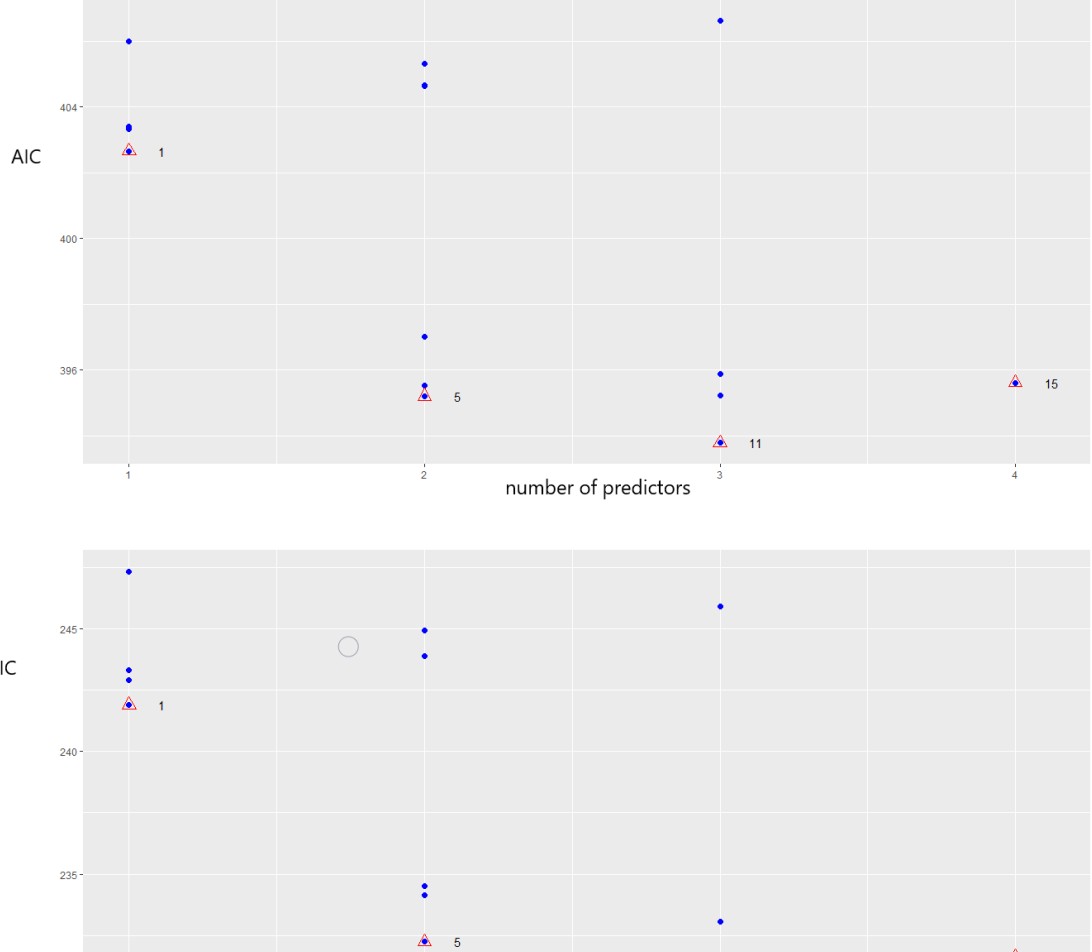

**Figure 3.** AIC values associated with Models (23) and (23), top and bottom panel. Each circle is associated to different combinations of predictors, while triangles highlight the best choice of predictors.

We tested the normality of the residuals, and the results in Table 7 confirmed the normality.

The results in Table 6 show that the excess mortality per 100k people mainly depended on the LRaG gap in 2020, while the excess mortality as a percentage of the annual baseline depended on the LRaG gap in 2019 and 2020.

We also looked for a direct linear dependence of the two excess mortalities on the LRaG gap in 2020, but these two univariate linear models did not work.

**Table 7.** Normality test for residuals of the multivariate linear regressions (22) and (23).

| Test | Statistic | *p*-Value |
|---|---|---|
| **Linear Regression (22)** | | |
| Shapiro–Wilk | 0.9814 | 0.8391 |
| Kolmogorov–Smirnov | 0.0630 | 0.9987 |
| Anderson–Darling | 0.1601 | 0.9427 |
| **Linear Regression (23)** | | |
| Test | Statistic | *p*-Value |
| Shapiro–Wilk | 0.9815 | 0.8429 |
| Kolmogorov–Smirnov | 0.0925 | 0.9236 |
| Anderson–Darling | 0.2581 | 0.6955 |

In contrast, there was a positive correlation between excess mortality and the LRaG gap in 2019. In fact, the following univariate linear models:

$$ME100k_{i,2021} = \beta_0 + \beta_1 dLRaG_{i,2019} + \epsilon_i, \tag{24}$$

$$MEAB_{i,2021} = \beta_0^* + \beta_1^* dLRaG_{i,2019} + \epsilon_i^*, \tag{25}$$

worked with *p*-values of F statistics of 0.078 and 0.0211, a multiple R-squared of 0.1 and 0.1649, and an adjusted R-squared of 0.07 and 0.1371, respectively. In Equation (24) and (25), as usual, $\epsilon_i$ and $\epsilon_i^*$ are normally distributed noise. The estimated coefficients are shown in Table 8.

**Table 8.** Multivariate linear regressions (24) and (25). Signif. codes: 0 "***", 0.001 "**", 0.01 "*", 0.05 ".", 0.1 " ", 1.

| Parameter | Estimate | Std. Error | *t* Value | Pr(>\|*t*\|) |
|---|---|---|---|---|
| **Linear Regression (24)** | | | | |
| (intercept) | 141.11 | 21.77 | 6.483 | $3.65 \times 10^{-7}$ *** |
| $dLRaG_{2019}$ | 44.92 | 24.60 | 1.826 | 0.0778 (·) |
| **Linear Regression (25)** | | | | |
| Parameter | Estimate | Std. Error | *t* Value | Pr(>\|*t*\|) |
| (intercept) | 13.077 | 1.766 | 7.406 | $2.98 \times 10^{-8}$ |
| $dLRaG_{2019}$ | 4.857 | 1.996 | 2.434 | 0.0211 (*) |

The results in Table 8 are interesting. They tell us that an increase of one year in the LRaG gap (i.e., people experienced a lower perceived age) predicts an increase in excess mortality per 100k people of about 45 deaths and an increase in the percentage of excess mortality with respect to the baseline of 4.85 percentage points. Thus, countries with lower LRaG gaps in 2019 are expected to show lower excess of mortality in 2020.

We further investigated this point, trying to understand whether the LRaG gap (i.e., dL-RaG indicator) corresponding to a given country is a measure of the country's resilience to a shock such as a pandemic.

We proceeded by ranking countries with respect to the dLRaG indicator in 2019 and with respect to excess mortality in 2020–2021 (i.e., EM100k and EMAB). We used the quartile ranking to define a measure of the association of the two variables. Specifically, we used Cramer's V measure, which is the preferred measure since its maximum value is 1 when

there is a very strong relationship and 0 when the categorical variables are independent. Cramer's V measure is defined as:

$$V_{EM100k,LRaG} = \sqrt{\frac{1}{3n} \sum_{i=1}^{4} \sum_{j=1}^{4} \frac{(n_{i,j} - \hat{n}_{i,j})^2}{\hat{n}_{i,j}}}, \tag{26}$$

where $n$ is the number of observations (i.e., $n = 32$), $n_{i,j}$ is the absolute frequency of the rank pair $(i, j)$, and $\hat{n}_{i,j}$ is the absolute frequency of the same pair under the assumption of independent variables. Table 9 shows the absolute frequencies.

**Table 9.** Bivariate absolute frequency distribution of the quartile ranking of the LRaG gap in 2019 and the excess of mortality per 100 k in 2020–21 (top panel) and excess mortality as a percentage of the annual baseline in 2020–2021 (bottom panel).

|  | **dLRaG_r1** | **dLRaG_r2** | **dLRaG_r3** | **dLRaG_r4** | **Row Tot.** |
|---|---|---|---|---|---|
| EMk100_r1 | 4 | 2 | 2 | 0 | 8 |
| EMk100_r2 | 1 | 2 | 2 | 3 | 8 |
| EMk100_r3 | 0 | 1 | 3 | 4 | 8 |
| EMk100_r4 | 3 | 3 | 1 | 1 | 8 |
| Col. Tot. | 8 | 8 | 8 | 8 | 32 |
|  | **dLRaG_r1** | **dLRaG_r2** | **dLRaG_r3** | **dLRaG_r4** | **Row Tot.** |
| EMAB_r1 | 4 | 1 | 3 | 0 | 8 |
| EMAB_r2 | 1 | 1 | 2 | 4 | 8 |
| EMAB_r3 | 1 | 3 | 2 | 2 | 8 |
| EMAB_r4 | 2 | 3 | 1 | 2 | 8 |
| Col. Tot. | 8 | 8 | 8 | 8 | 32 |

Table 9 shows that the quartile rankings generated by EM100k and the LRaG gap (i.e., dLRaG indicator) classified 10 units out of 32 (percentage 32%) with the same rank, while the quartile rankings generated by EMBA and the LRaG gap assigned the same rank to nine units out of 32 (percentage 28%). We further investigated the relationship between the LRaG gap and excess mortality by computing the absolute difference of the two ranking variables. Table 10 shows the distribution of the absolute difference of the quartile ranking of the LRaG gap in 2019 and excess mortality per 100k people in 2020–2021 (top panel) and excess mortality as a percentage of the annual baseline in 2020–2021 (bottom panel).

**Table 10.** Distribution of the absolute difference of the quartile ranking of the LRaG gap in 2019 and excess mortality per 100k people in 2020–2021 (top panel) and excess mortality as a percentage of the annual baseline in 2020–2021 (bottom panel).

| $\lvert Rank\, EMk100_{2020-21} - Rank\, dLRaG_{2019} \rvert$ | 0 | 1 | 2 | 3 |
|---|---|---|---|---|
| n. units | 10 | 11 | 8 | 3 |
| $\lvert Rank\, EMAB_{2020-21} - Rank\, dLRaG_{2019} \rvert$ | 0 | 1 | 2 | 3 |
| n. units | 9 | 10 | 11 | 2 |

Table 10 provides empirical evidence that the excess mortality per 100k in 2020–2021 was closely related to the LRaG gap in 2019 since the quartile ranking classification shared 22 out of 32 cases under the median, i.e., a percentage of 68.75%, while the percentage of shared classification reduced to 59.37% in the case of excess mortality as a percentage of the annual baseline.

The computed value of Cramer's V was $V_{EM100k,LRaG} = 0.35$, indicating that the variables were not independent. We also measured the robustness of the result using a Cramer test (R packages "cramer", [17]) for a two-sample problem to test the equality of

two-sample distributions. To calculate the critical value, Monte Carlo bootstrap methods and eigenvalue methods were used. This test also works with small sample sizes.

In detail, we used the Cramer test to compare the distribution of the two categorical variables: quartile ranking obtained with EM100k in 2020–2021 and dLRaG in 2019. Based on 1000 ordinary bootstrap replicates, the critical value for a confidence level of 95 % was 1.828, so the hypothesis "EM100k is distributed as dLRaG" was accepted with estimated *p*-value = 0.998.

We repeated the experiment, analyzing whether the indicator EMAB 2020-21 was associated with the LRaG 2019, that is we computed:

$$V_{EMAB,LRaG} = \sqrt{\frac{1}{3n} \sum_{i=1}^{4} \sum_{j=1}^{4} \frac{(n_{i,j} - \hat{n}_{i,j})^2}{\hat{n}_{i,j}}}. \tag{27}$$

Cramer's V was equal to 0.32, that is slightly lower than the previous one, while the Cramer test confirmed that there was no empirical evidence to reject the null hypothesis that the categorical variable of the dLRaG quartile was distributed as the variable of the EMAB-indicator quartile since the *p*-value was 0.996. In Appendix A, we report the quartile ranking in Table A3.

The fact that the dLRaG-2019 quartiles and the quartiles of the mortality excess were drawn from the same distribution suggests that countries with a low dLRaG-2019 rank are resilient to health shocks in that they were associated with low excess mortality in 2020–2021.

We conclude by establishing whether the quartile variable defined by the health expenditure in 2019 was associated with the dLRaG in 2017 and 2020. Cramer's V measure was equal to 0.29 and 0.31, respectively, while the distribution of the absolute value of the differences in the quartile variables is shown in Table 11.

**Table 11.** Distribution of the absolute difference of the quartile ranking of health expenditure in 2019 and the LRagG gap in 2017 (top panel) and in 2020 (bottom panel).

| $\lvert Rank\ HE_{2019} - Rank\ dLRaG_{2017} \rvert$ | 0 | 1 | 2 | 3 |
|---|---|---|---|---|
| n. units | 10 | 11 | 8 | 3 |
| $\lvert Rank\ HE_{2019} - Rank\ dLRaG_{2020} \rvert$ | 0 | 1 | 2 | 3 |
| n. units | 7 | 12 | 7 | 6 |

Table 11 and Figures 4 and 5 show that countries such as Bulgaria, Israel, Italy, Russia, Slovenia, and Sweden experienced a decrease in the LRaG gap from 2017 to 2020, i.e., an improvement in biological age, and the health expenditure in these countries was only in the first quartile, except for Israel and Italy, which were in the third quartile.

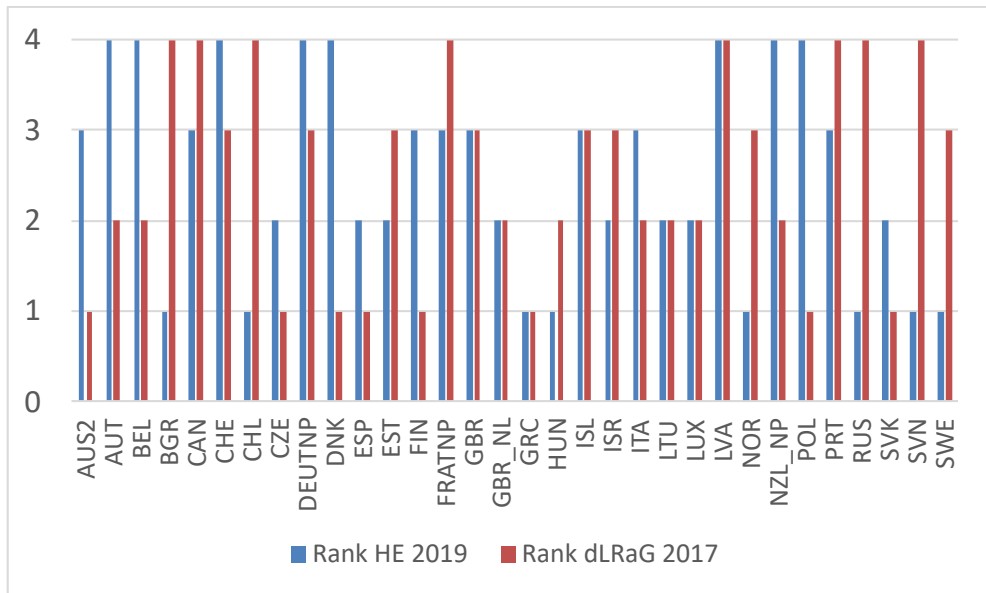

**Figure 4.** Country ranking by health expenditure in 2019 (i.e., HE 2019) and by LRaG gap in 2017. Countries on the horizontal axis are sorted alphabetically.

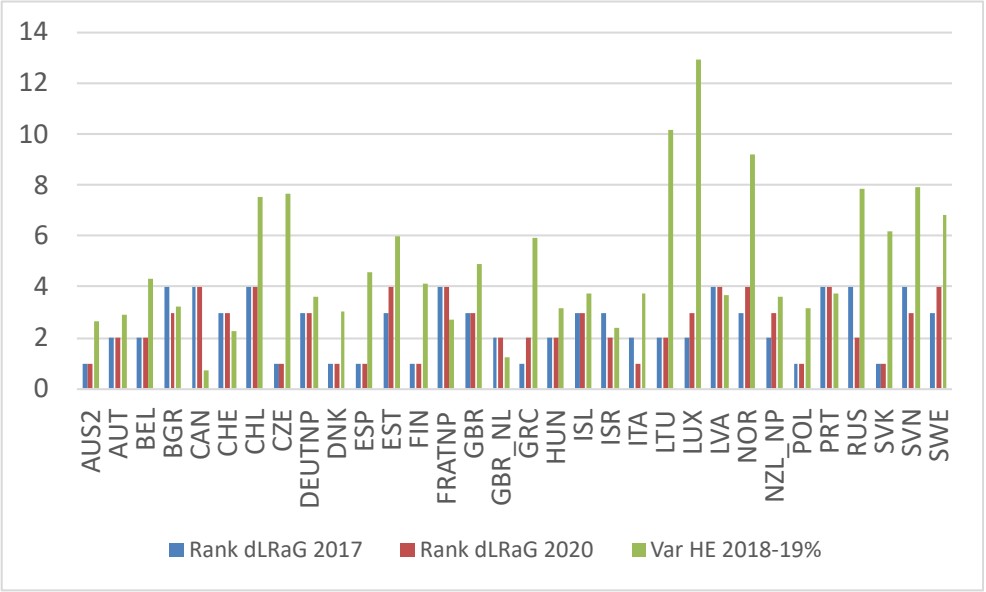

**Figure 5.** Country ranking by LRaG gap in 2017 and 2020 and the variation of health expenditure in 2018–2019 as a percentage. Countries on the horizontal axis are sorted alphabetically.

In contrast to the above-mentioned countries, Estonia, Greece, Luxembourg, Norway, New Zealand, and Sweden experienced a decrease in the LRaG gap from 2017–2020, despite a large increase in health expenditure from 2018 to 2019, but with health expenditure in the first or second quartile, except for New Zealand.

Interestingly, countries with unchanged LRaG gap rankings were those with large expenditures; see, for example, Australia, Austria, Belgium, Canada, Chile, Czechia, Denmark, Finland, France, Hungary, and Latvia.

Remarkably, countries with a higher health expenditure (HE quartile ranks 3 and 4) were those with a higher quality of life (i.e., lower values of the quartile rank of the LRaG gap in 2020).

We conclude this section with a look at the per-country ranking of excess mortality per 100k people in 2020–2021 and the LRaG gap in 2019, as displayed in Figure 6. In this figure, the countries on the horizontal axis are sorted by LRaG gap rank.

Looking at Figure 6, we see that excess mortality was frequently ranked as the LRaG gap plus or minus one. For four countries—Czechia, Italy, Slovakia, Hungary, and Lithuania —we observed large values of excess mortality and good quality of life (biological age less than chronological age). This could be explained by elderly people with a good quality of life who were strongly affected by a COVID-19 wave and strained the healthcare system. In contrast to these countries, we found countries with a large LRaG gap but low excess mortality, such as Luxembourg, Norway, and Sweden, countries that had a low health expenditure per capita. This is a very preliminary interpretation that deserves further investigation.

We note that countries with a constant quartile for the LRaG gap from 2017 to 2020 were those with the lowest excess mortality (i.e., ranks 1 or 2).

Hence, the constant dynamics of the LRaG gap over time seems to be predictive of the resilience of the country to a health shock when expressed as low excess mortality.

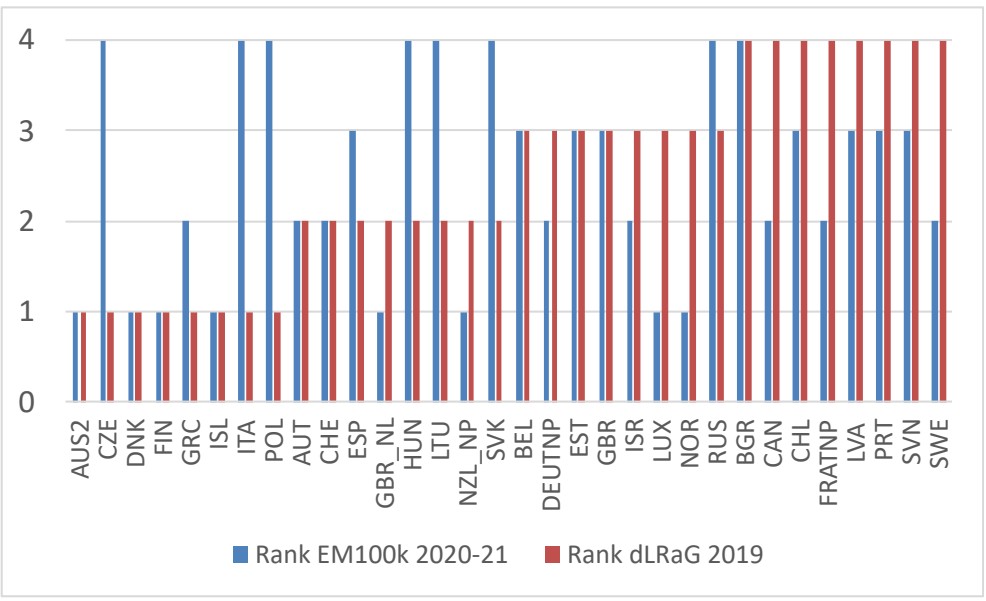

**Figure 6.** Country ranking by excess mortality per 100k people (i.e., EM100k) in 2020–2021 and by LRaG gap in 2019.

## 4. Discussion

We introduced the weekly LRaG-age estimator, i.e., the estimator of biological age, using the new dataset of short-term mortality fluctuations. The difference between biological and chronological age, i.e., the dLRaG indicator, was shown to be related to health expenditure, the excess mortality per 100k people, and excess mortality as a percentage of the annual baseline. The constant dynamics of the quartile rank of the LRaG gap seems to be predictive of the resilience of a country. The LRaG was analyzed by Milevsky in 2020 [10] using annual data from the human mortality dataset. In [10], the connection between LRaG age and biological age was detailed, along with applications to pension and retirement policies.

**Author Contributions:** Conceptualization, M.C.R. and G.P.; methodology, M.C.R. and G.P.; validation, M.C.R., G.P., A.R. and A.S.; writing—original draft preparation, M.C.R.; writing—review and editing, G.P. and A.S.; visualization, M.C.R. and G.P.; supervision, M.C.R.; project administration, M.C.R. All authors have read and agreed to the published version of the manuscript.

**Funding:** This research received no external funding.

**Institutional Review Board Statement:** Not applicable.

**Informed Consent Statement:** Not applicable.

**Data Availability Statement:** The data used in this work can be freely downloaded at https://www.mortality.org/ (accessed on 21 February 2022).

**Conflicts of Interest:** The authors declare no conflict of interest.

## Abbreviations

The following abbreviations are used in this manuscript:

| | |
|---|---|
| L-RaG | longevity-risk-adjusted global |
| GM | Gompertz–Makeham |
| TMR | total mortality rate |
| wTMR | weekly total mortality rate |
| INMR | initial natural mortality rate |
| ME 100k | mortality excess per 100k |
| MEAB | mortality excess as a percentage of the annual baseline |
| HE | health expenditure |

## Appendix A

This Appendix contains some tables reporting the results.

**Table A1.** Country code meaning.

| Country Code | Country Name | Country Code | Country Name |
|---|---|---|---|
| AUS | Australia | LVA | Latvia |
| AUT | Austria | LTU | Lithuania |
| BLR | Belarus | LUX | Luxembourg |
| BEL | Belgium | NLD | Netherlands |
| BGR | Bulgaria | NZL_NP | New Zealand Total population |
| CAN | Canada | NZL_MA | New Zealand Maori |
| CHL | Chile | NZL_NM | New Zealand Non-Maori |
| HRV | Croatia | NOR | Norway |
| CZE | Czechia | POL | Poland |
| DNK | Denmark | PRT | Portugal |
| EST | Estonia | KOR | Republic of Korea |
| FIN | Finland | RUS | Russia |
| FRATNP | France Total population | SVK | Slovakia |
| FRACNP | France Civilian population | SVN | Slovenia |
| DEUTNP | Germany Total population | ESP | Spain |
| DEUTE | Germany East Germany | SWE | Sweden |
| DEUTW | Germany West Germany | CHE | Switzerland |
| GRC | Greece | TWN | Taiwan |
| HKG | Hong Kong | GBR_NP | U.K. United Kingdom Total Population |
| HUN | Hungary | GBRTENW | U.K. England and Wales Total Population |
| ISL | Iceland | GBRCENW | U.K. England and Wales Civilian Population |
| IRL | Ireland | GBR_SCO | U.K. Scotland |
| ISR | Israel | GBR_NIR | U.K. Northern Ireland |
| ITA | Italy | USA | U.S.A. |

**Table A2.** Data used in the linear regressions.

| Country | ME 100k | MEAB | HE 2019 | D2017 | D2018 | D2019 | D2020 |
|---|---|---|---|---|---|---|---|
| AUS2 | −14.40 | −2.50 | 4919.24 | −2.30 | −1.80 | −2.29 | −1.23 |
| AUT | 108.90 | 11.70 | 5705.10 | −0.31 | −0.52 | −0.24 | −0.16 |
| BEL | 138.70 | 14.50 | 5458.40 | 0.06 | 0.12 | 0.10 | −0.15 |
| BGR | 457.50 | 29.00 | 1842.05 | 0.72 | 0.64 | 0.62 | 0.21 |
| CAN | 40.00 | 5.10 | 5370.44 | 1.19 | 0.99 | 1.14 | 1.26 |
| CHE | 99.70 | 12.60 | 7138.06 | 0.14 | 0.20 | −0.00 | 0.23 |

**Table A2.** *Cont.*

| Country | ME 100k | MEAB | HE 2019 | D2017 | D2018 | D2019 | D2020 |
|---------|---------|------|---------|-------|-------|-------|-------|
| CHL | 158.50 | 26.90 | 2291.46 | 1.94 | 1.61 | 2.00 | 0.98 |
| CZE | 323.60 | 30.10 | 3417.49 | −0.99 | −0.81 | −0.87 | −0.85 |
| DEUTNP | 47.10 | 4.00 | 6518.00 | 0.07 | 0.04 | 0.12 | 0.21 |
| DNK | −10.80 | −1.10 | 5477.57 | −0.57 | −0.44 | −0.69 | −0.26 |
| ESP | 186.20 | 20.30 | 3600.28 | −0.48 | −0.36 | −0.30 | −0.30 |
| EST | 137.90 | 11.50 | 2507.07 | 0.48 | 0.50 | 0.17 | 0.52 |
| FIN | 7.40 | 0.70 | 4558.54 | −1.18 | −1.31 | −1.42 | −0.74 |
| FRATNP | 109.60 | 11.60 | 5274.26 | 0.84 | 0.75 | 0.95 | 0.51 |
| GBR | 160.70 | 17.70 | 4500.14 | 0.22 | 0.18 | 0.35 | 0.11 |
| GBR_NL | 28.50 | 4.50 | 2318.96 | −0.21 | −0.07 | −0.14 | −0.08 |
| GRC | 72.40 | 5.90 | 2014.20 | −0.47 | −0.37 | −0.53 | −0.23 |
| HUN | 243.60 | 17.80 | 2169.77 | −0.26 | −0.53 | −0.10 | −0.21 |
| ISL | −4.50 | −0.70 | 4540.76 | 0.14 | −0.21 | −1.60 | 0.08 |
| ISR | 56.00 | 10.40 | 2903.41 | 0.14 | 0.05 | 0.07 | −0.19 |
| ITA | 206.30 | 19.10 | 3653.40 | −0.42 | −0.14 | −0.60 | −0.45 |
| LTU | 350.70 | 24.90 | 3406.26 | −0.30 | −0.23 | −0.22 | −0.14 |
| LUX | 31.00 | 4.30 | 2727.19 | −0.02 | 0.16 | 0.31 | −0.06 |
| LVA | 158.10 | 10.40 | 5414.48 | 0.67 | 0.89 | 0.72 | 0.35 |
| NOR | −28.20 | −3.70 | 2039.22 | 0.38 | −0.00 | 0.18 | 0.38 |
| NZL_NP | −40.00 | −5.40 | 5739.20 | −0.29 | −0.35 | −0.20 | 0.02 |
| POL | 309.30 | 27.60 | 6744.62 | −1.42 | −1.26 | −1.18 | −0.88 |
| PRT | 184.90 | 16.30 | 4211.85 | 0.58 | 0.58 | 1.23 | 0.78 |
| RUS | 339.80 | 28.20 | 2289.31 | 0.48 | 0.31 | 0.30 | −0.08 |
| SVK | 305.40 | 30.50 | 3347.43 | −0.52 | −0.06 | −0.15 | −0.27 |
| SVN | 178.70 | 17.40 | 1850.26 | 1.28 | 0.73 | 0.59 | 0.04 |
| SWE | 88.10 | 9.70 | 2189.05 | 0.35 | 0.53 | 0.87 | 0.47 |

**Table A3.** Quartile ranking with respect to excess mortality per 100k people in 2020–2021, the dLRaG indicator in 2019, and excess mortality per annual baseline.

| Country | Rank EM100k 2020–2021 | Rank dLRaG 2019 | Rank EMAB 2020–2021 |
|---------|----------------------|-----------------|---------------------|
| AUS2 | 1 | 1 | 1 |
| AUT | 2 | 2 | 3 |
| BEL | 3 | 3 | 3 |
| BGR | 4 | 4 | 4 |
| CAN | 2 | 4 | 2 |
| CHE | 2 | 2 | 3 |
| CHL | 3 | 4 | 4 |
| CZE | 4 | 1 | 4 |
| DEUTNP | 2 | 3 | 1 |
| DNK | 1 | 1 | 1 |
| ESP | 3 | 2 | 4 |
| EST | 3 | 3 | 2 |
| FIN | 1 | 1 | 1 |
| FRATNP | 2 | 4 | 2 |
| GBR | 3 | 3 | 3 |
| GBR_NL | 1 | 2 | 2 |
| GRC | 2 | 1 | 2 |
| HUN | 4 | 2 | 3 |
| ISL | 1 | 1 | 1 |
| ISR | 2 | 3 | 2 |
| ITA | 4 | 1 | 3 |
| LTU | 4 | 2 | 4 |
| LUX | 1 | 3 | 1 |
| LVA | 3 | 4 | 2 |

**Table A3.** *Cont.*

| Country | Rank EM100k 2020–2021 | Rank dLRaG 2019 | Rank EMAB 2020–2021 |
|---------|-----------------------|-----------------|---------------------|
| NOR | 1 | 3 | 1 |
| NZL_NP | 1 | 2 | 1 |
| POL | 4 | 1 | 4 |
| PRT | 3 | 4 | 3 |
| RUS | 4 | 3 | 4 |
| SVK | 4 | 2 | 4 |
| SVN | 3 | 4 | 3 |
| SWE | 2 | 4 | 2 |

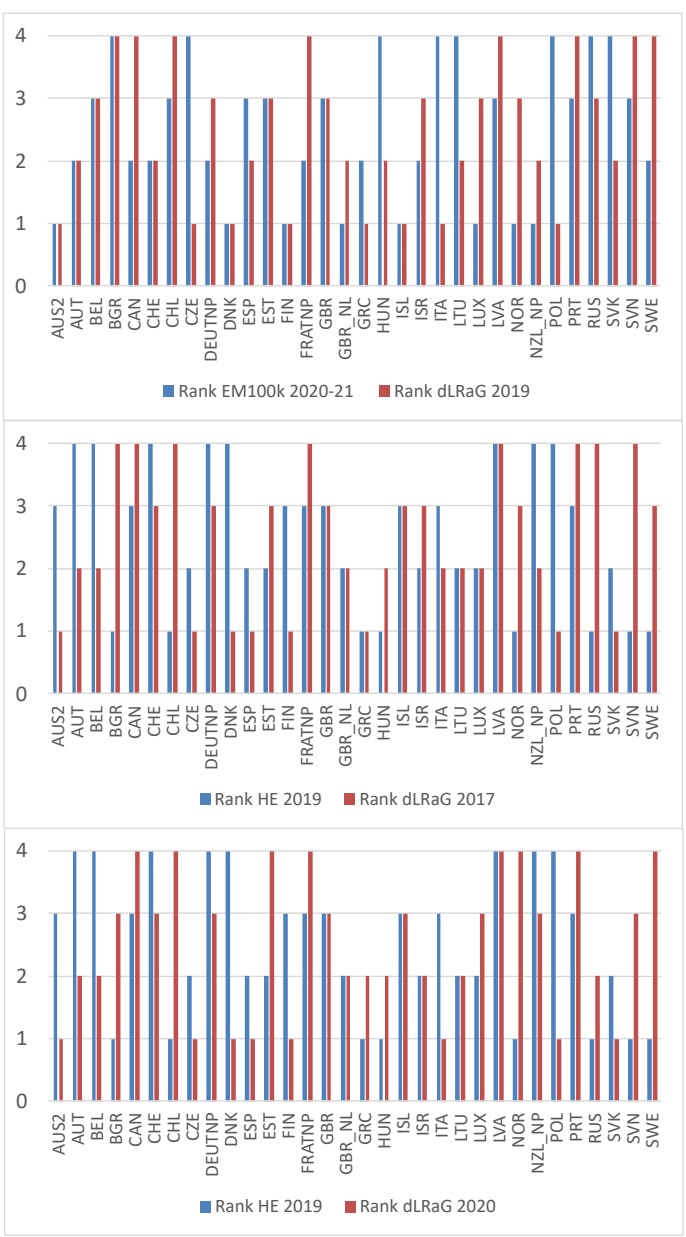

**Figure A1.** Country ranking by excess mortality per 100k people (i.e., EM100k) in 2020–2021 and LRaG gap in 2019 (upper panel); by health expenditure in 2019 and LRaG gap in 2017 (middle panel); by health expenditure in 2019 and LRaG gap in 2020 (lower panel). Countries on the horizontal axis are sorted alphabetically.

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
