# Peer review of "Short-Term Mortality Fluctuations and Longevity Risk-Adjusted Age: Learning the Resilience of a Country to a Health Shock"

_computation, doi:10.3390/computation10040047_

Round 1
Reviewer 1 Report
This was an outstanding and well polished paper providing insight into the consequences of COVID. Aside from some small typos that do not affect the quality of the work, the single criticism I have of this paper is that the color scheme in Figure 2 is not suitable for publication. The reader cannot reasonably differentiate different shades of that particular blue, especially considering how small some of the represented nations are in the sub-figures. Since restricting the sub-figures to those nations with data will not likely improve the situation, I would recommend a multi-chromatic color scheme so that relative differences in values can be appreciate by readers. The typos I mentioned earlier appear in line 78 (naming the differenced indicator, line 182 (some strikethrough text from prior editing needs to be removed), and line 190 (new paragraph requires indentation).
Congratulations on a great work!
Author Response
Thank you very much for your positive feedback! We tried to improve the paper based on your opinion.
Reviewer 2 Report
This paper introduces biological age estimator (LRaG) and the difference
dLRaG = LRaG - [chronological age] is used for further analysis.
(1) dLRaG is calculated for year 2017, 2018, 2019 and 2020.
(2) Multiple linear regressions are fitted with response ME100k, MEAB, and covariates dLRaG for 2017, 2018, 2019 and 2020.
(3)Since d LRaG for 2017, 2018, 2019 and 2020 are highly correlated, only dLRaG for 2019 is kept in the model.
Applying just multiple linear regression seem too naive. The procedure waisting the information. Maybe consider longitudinal data analysis.
Author Response
Dear colleague,
In order to justify the fact that we have eliminated dLRag for 2018 in the model, we added this comment:
“Since the higher values of Variance Inflation Ratio (VIF) associated with variables dLRaG2017 and dLRaG2018, 17.14 and 22.27 according to both models (22)
and (23), we eliminate dLRaG2018 from the models. Moreover, Figure 3 shows akaike (AIC) values for stepwise linear regression based on all possible combinations of predictors.
Models (22) and (23) reach the minimum value of AIC only if the three predictors are included in the analysis (AIC=393 vs AIC=230).”
The minimum value of AIC only if the three predictors are included in the analysis (dLRaG2017, dLRaG2019, dLRaG2020). Only dLRaG for 2019 is kept in the model looking at the significance of the coefficients in Table 6.
Multiple linear regression allows us to estimate the impact of dLRaG on ME100ki,2021 and MEABi,2021 since we are interesting in the prediction not longitudinal analysis in fact values of ME100 and MEAB for 2019 and 2020 are missing in the models (22) and (23).
Reviewer 3 Report
The authors proposed a weekly LRaG-age estimator using the new dataset of short-term mortality fluctuations. The difference between biological and chronological age, is shown to be related to health expenditure, the excess mortality per 100k people, and excess mortality as a percentage of annual baseline. The results are interesting and analysis is technically correct. Some few minor comments should be addressed:
- The literature review is not comprehensive and introduction does not reflect the main core of research effectively.
- Conclusion does not highlight the outputs.
- Needs to add more figures to give a better graphical demonstration of the data analysis
Author Response
Thank you for the responce! We have tried to improve the paper based on your comments.